# Compression Neuropathies of the Upper Extremity: A Review

Carter J. Boyd [1,*], Nikhi P. Singh [2], Joseph X. Robin [3] and Sheel Sharma [1]

[1] Hansjörg Wyss Department of Plastic Surgery, NYU Langone, 564 1st Avenue, New York, NY 10016, USA; Sheel.Sharma@nyulangone.org

[2] School of Medicine, University of Alabama at Birmingham, Birmingham, AL 35233, USA; nsingh@uab.edu

[3] Department of Orthopaedic Surgery, NYU Langone, New York, NY 10010, USA; josephxrobin@gmail.com

* Correspondence: carterjosephboyd@gmail.com; Tel.: +1-318-780-1049

**Abstract:** Compressive neuropathies of the forearm are common and involve structures innervated by the median, ulnar, and radial nerves. A thorough patient history, occupational history, and physical examination can aid diagnosis. Electromyography, X-ray, and Magnetic Resonance Imaging may prove useful in select syndromes. Generally, first line therapy of all compressive neuropathies consists of activity modification, rest, splinting, and non-steroidal anti-inflammatory drugs. Many patients experience improvement with conservative measures. For those lacking adequate response, steroid injections may improve symptoms. Surgical release is the last line therapy and has varied outcomes depending on the compression. Carpal Tunnel syndrome (CTS) is the most common, followed by ulnar tunnel syndrome. Open and endoscopic CTS release appear to have similar outcomes. Endoscopic release appears to incur decreased cost baring a low rate of complications, although this is debated in the literature. Additional syndromes of median nerve compression include pronator syndrome (PS), anterior interosseous syndrome, and ligament of Struthers syndrome. Ulnar nerve compressive neuropathies include cubital tunnel syndrome and Guyon's canal. Radial nerve compressive neuropathies include radial tunnel syndrome and Wartenberg's syndrome. The goal of this review is to provide all clinicians with guidance on diagnosis and treatment of commonly encountered compressive neuropathies of the forearm.

**Keywords:** compressive neuropathies; carpal tunnel syndrome; anterior interosseous syndrome; cubital tunnel syndrome; pronator syndrome; ligament of Struthers syndrome; ulnar tunnel syndrome; radial tunnel syndrome; Wartenberg syndrome; carpal tunnel release; endoscopic; upper extremity

## 1. Introduction

Compression neuropathies of the upper extremity are a common problem encountered among clinicians. Early symptom recognition, thorough diagnostic workup, and appropriate treatment is critical for these patients to prevent further complications and potential long-term sequalae. Etiologies such as trauma, entrapment, compartment syndrome, and edema can cause acute nerve compression and therefore, acute nerve ischemia. Chronic nerve compression may result in damage to the myelin sheath requiring regeneration which can take up to 12 weeks [1–3]. In some cases, peripheral nerve damage secondary to extrinsic compression can result in permanent motor and/or sensory deficits. Severe cases of compression, such as crush injuries, may result in Wallerian degeneration in which the nerve regrows approximately 1 inch a month [3]. Within, we aim to provide the reader with a compressive understanding of common compressive neuropathies, aiding clinicians in diagnosis and treatment. Content included in this review was found through an extensive literature review performed using key terms: "compressive neuropathies"; "carpal tunnel syndrome"; "; "carpal tunnel release"; "endoscopic carpal tunnel release"; "open carpal tunnel release"; "carpal tunnel release cost"; "anterior interosseous syndrome"; "cubital tunnel syndrome"; "pronator syndrome"; "ligament of Struthers syndrome"; "ulnar tunnel syndrome"; "Guyon's canal"; "radial tunnel syndrome"; "Wartenberg syndrome. The most relevant articles were used as sources.

Three major nerves course through the forearm on their way to innervating the distal extremity—the median, ulnar, and radial nerves. Each can be affected in compression syndromes. Understanding the muscular anatomy, innervation, and sensory distribution of the upper extremity is critical for diagnosing and treating compressive neuropathies.

Initial assessment of patients with suspected compression neuropathies should include a thorough history, taking note of any underlying systemic diseases or conditions as well as past medical or surgical history that could alert you to the etiology of their symptoms. Identification of the patient's hand dominance and occupation is key, as these factors will likely impact the patient's ability to cooperate with different treatment modalities. In addition, evaluation for more proximal injuries, notably cervical radiculopathy should be emphasized. Understanding the etiology of the injury is important, as patients presenting with work related injuries have been shown to have worse outcomes relative to other patient populations [4]. Additional workup includes a complete physical exam of the neck, shoulder, and upper extremities and specific tests based on the patient's clinical narrative, such as the Tinel's sign, Phalen's maneuver, and other provocative tests. Electromyography (EMG) or nerve conduction studies can be useful but are not always necessary for diagnosis, as these studies are often not indicated and are highly operator dependent [5]. Imaging modalities such as X-ray or Magnetic Resonance Imaging (MRI) are often beneficial in patients with an acute or remote history of trauma, fracture, or oncologic disease in the affected extremity. Table 1 demonstrates a typical treatment pathway for these patients.

**Table 1.** Typical generalized treatment pathway for patients with a compressive neuropathy of the upper extremity.

| Level | Treatment | Benefit |
|---|---|---|
| Initial Management | | |
| Baseline | Conservative management, including rest and splinting of affected extremity. Physical therapy may also aid in reduction of symptoms. | Decreases compression and irritation of affected nerve. |
| Medication if needed | Non-steroidal anti-inflammatory agents (NSAIDs) | Assists with pain management and decreases inflammation. |
| Optimization of comorbid conditions | Proper management of existing conditions which may include smoking cessation, weight loss, glycemic control, and regular exercise. | Holistically addresses the patient and may prevent the patient progressing to more involved and invasive treatments. |
| If Initial Management Fails | | |
| Medication escalation | Steroid injections. | Helps with pain management, inflammation, and diagnosis. |
| Final treatment | Surgery | Definitive decompression of nerve. |

## 2. Compression Syndromes

### 2.1. Median Nerve

#### 2.1.1. Carpal Tunnel Syndrome

Carpal tunnel syndrome (CTS) is the most common peripheral compressive mononeuropathy affecting 276 per 100,000 persons [6,7]. CTS is due to compression of the median nerve at the wrist within the carpal tunnel which is bound by carpal bones and the transverse carpal ligament. A comprehensive summary of the compressive neuropathies of the median nerve is provided in Table 2. Structures within the carpal tunnel include the median nerve, four flexor digitorum superficialis (FDS) tendons, four flexor digitorum profundus (FDP) tendons, and the flexor policus longus (FPL) tendon. The FDS tendons of the middle and ring fingers course volar within the carpal tunnel relative to the index and small finger FDS tendons. The lumbricals, which are attached to FDP tendons, may enter the carpal tunnel with finger flexion and can cause effort-associated CTS, especially

in those with muscular hands [8]. Risk factors for CTS include any condition exacerbating edema, inflammation, hormonal changes, and repetitive manual activity [9].

**Table 2.** Summary of median nerve compressive neuropathies.

| Median Nerve | |
|---|---|
| **Carpal Tunnel Syndrome** | |
| Compression Site | Carpal tunnel containing median nerve, four flexor digitorum superficialis (FDS) tendons, four flexor digitorum profundus (FDP) tendons, and the flexor policus longus (FPL) tendon. |
| Sensation Deficit | Intermittent, nocturnal paresthesia's and dysesthesias in thumb, index finger, middle finger, and medial aspect of ring finger. |
| Diagnosis | Presence of sensory defects and thenar atrophy. Advanced disease may produce thenar atrophy. Positive Tinel and Phalen sign. Electromyography (EMG) can be used to confirm diagnosis. |
| **Anterior Interosseous Syndrome** | |
| Compression Site | Usually in forearm within fibrous arch of FDS, less often by Gantzer muscle (accessory FPL). |
| Sensation Deficit | No sensory deficits. |
| Symptoms | Motor weakness of FPL and FDP of index and middle fingers. |
| Diagnosis | Positive Tinel sign, "Kiloh-Nevin Sign", pinch maneuver. EMG with sharp waves, fibrillations, and abnormal latencies across affected muscles |
| **Pronator Syndrome** | |
| Compression Site | Between two heads of the pronator teres muscles or proximal arch of the FDS. |
| Symptoms | Volar forearm pain and paresthesia in median nerve distribution. |
| Diagnosis | Pain on compression of proximal volar forearm. EMG studies rule out other syndromes. |
| **Ligament of Struthers Syndrome** | |
| Caused by rare anatomical accessory fibrous band between the supracondylar process of the humerus and the medial humeral epicondyle. Causes pain, weakness, and sensory defects in median nerve distribution. | |

### 2.1.2. Diagnosis and Assessment

The gold standard for diagnosis of CTS is through a thorough history and physical examination. Progression of this disease is well established and typically commences with intermittent, nocturnal paresthesia's and dysesthesias [9]. As the syndrome progresses, these abnormal feelings increase in frequency and start to occur while the patient is awake. Patients may report numbness, tingling, and motor weakness instead of pain. The markers for disease progression are sensory deficits and thenar muscle atrophy. Electrophysiological assessment through EMG is highly sensitive in detecting CTS and are diagnostically useful in conjunction with clinical findings [9,10]. EMG findings indicative of CTS include fibrillation potentials and positive sharp waves. These studies can help localize the site of compression when clinical findings are ambiguous.

### 2.1.3. Treatment

First line treatment for CTS is based off of severity of disease. For those without sensory loss or thenar atrophy, conservative treatment options such as splinting, anti-inflammatory agents, activity modification, and steroid injections are indicated. Steroid therapy has been proven to have similar short-term outcomes as surgery for patients with CTS, however outcomes past a year are superior for patients undergoing release surgery [11,12]. For patients who fail non-operative treatment or for those with worsening disease, surgical carpal tunnel release can be offered via an open or endoscopic approach. Both approaches are reliable and associated with low risk for complications such as damage to nerves, arteries or tendons [9,13]. Complex regional pain syndrome is a severe complication following carpal tunnel release affecting 2.5–8.3% of patients [14,15]. Incomplete release requiring revision and transient neuropraxia is a more common complication [10].

A 2019 study by Hubbard et al. found an estimated 4.8 billion dollar a year economic loss due to CTS and that treatment with surgery provided a 1.6 billion dollar benefit a year in the US [16]. Schrier et al. found of patients undergoing release surgery for CTS, that patients reporting more satisfactory experiences with their healthcare provider and health system also reported better outcomes following surgery [17]. In a cohort of 500 patients undergoing over 600 carpal tunnel releases, Kaltenborn et al. found that aspirin could be taken safely throughout the perioperative period for release surgery as there were no differences observed in patients taking aspirin [18]. A 2020 systematic review by Olaiya et al. with 765 patients and 866 hands found no clinical benefit to use of a tourniquet in carpal tunnel release surgery [19]. Operative time was on average 2 min faster in those who used a tourniquet, however postoperative pain scores in those patients were significantly higher compared to patients who did not have a tourniquet [19].

### 2.1.4. Open vs. Endoscopic

Extensive discussion exists regarding open versus endoscopic carpal tunnel release (CTR) [13,20–26]. Chen et al. in a meta-analysis of 15 randomized controlled trials totaling nearly 1600 hands found similar CTS symptom relief between patients who had open and endoscopic CTR procedures [20]. Endoscopic surgical approach in these patients was associated with a quicker recovery, an earlier return to work, and less severe complications than open release [20]. Michelotti et al. in a cohort of 25 patients with bilateral CTS who were treated on one hand with open and the other endoscopic surgery, found higher patient reported satisfaction with endoscopic surgery [24]. However, a prospective randomized trial by Atroshi et al. published in JAMA, found no difference in patient reported long-term outcomes between open and endoscopic surgery [21].

There is extensive discussion in the literature regarding the cost of open and endoscopic techniques (Table 3) [27]. A 1998 study by Chung and et al. found endoscopic release to be more cost effective and concluded that younger patients may benefit from this technique [28]. This finding was contradicted by Koehler et al. in a 2019 study, which found that endoscopic was 44% more expensive than open, largely due to cost of endoscopic instruments and the increased time associated with training surgeons and trainees endoscopic technique [27]. By contrast, Vasen and associates developed a decision making tree in their analysis and found that for the average patient, endoscopic release was more cost effective. Additional studies have found decreased healthcare system costs and increased profit margins for surgeons performing carpal tunnel release in ambulatory settings versus a traditional operating room. This is likely due to costs attributable to operating room inefficiencies, including time and costs associated with anesthesia [29,30].

**Table 3.** Cost comparison for open vs. endoscopic carpal tunnel release surgery.

| Carpal Tunnel Release Surgery Cost Findings | |
| --- | --- |
| **Author:** | **Finding:** |
| Chung et al. (1998) [28] | Endoscopic release is cost effective baring low rate of median nerve damage. |
| Koehler et al. (2019) [27] | Open release incurs a lower cost than endoscopic. |
| Vasen et al. (1999) [25] | Endoscopic release is cost effect with low complication rate. |
| Chatterjee et al. (2011) [29] | Release surgery performed in the clinic has less cost than compared to release in OR. |

Anterior Interosseous Syndrome.

Carpal tunnel release surgery is one of the most common procedures performed in plastic and orthopedic surgery, and in a 2020 study by Vargas and colleagues, trainees reported the highest levels of operative autonomy with open carpal tunnel release [31]. Preoperative preparation with a surgical video has been shown to reduce resident error and increase operative confidence [32]. CTS continues to be a highly studied and discussed topic within hand surgery given its prevalence and impact on patients.

### 2.1.5. Etiology

Anterior interosseous nerve syndrome (AINS) is a proximal median nerve isolated motor neuropathy. The anterior interosseous nerve (AIN) branch of the median nerves passes deep to the fibrous arch of FDS and provides motor innervation to the FPL, FDP of the index and middle finger, and the pronator quadratus muscles. It carries no cutaneous sensory innervation, therefore patients with an isolated AINS will lack sensory deficits. The etiology of AINS is often spontaneous, iatrogenic, or secondary to an upper extremity trauma (Figure 1). Cases of AIN caused by an accessory FPL, the Gantzer muscle, have been described [33,34]. Spontaneous cases of AINS may be due to an inflammatory neuritis caused by a viral illness, immunization reaction, peripartum periods, or strenuous exercise (Figure 1) [35]. Stutz reported a prodromal phase of flu-like symptoms and diffuse forearm pain preceding the development of motor weakness [36,37]. Recent studies have suggested that AINS should be reclassified as a transient idiopathic nerve dysfunction syndrome such as Parsonage-Turner or a brachial plexus neuritis instead of its current classification as a compressive neuropathy [36,38]. However, evidence to support this claim lacks adequate evidence.

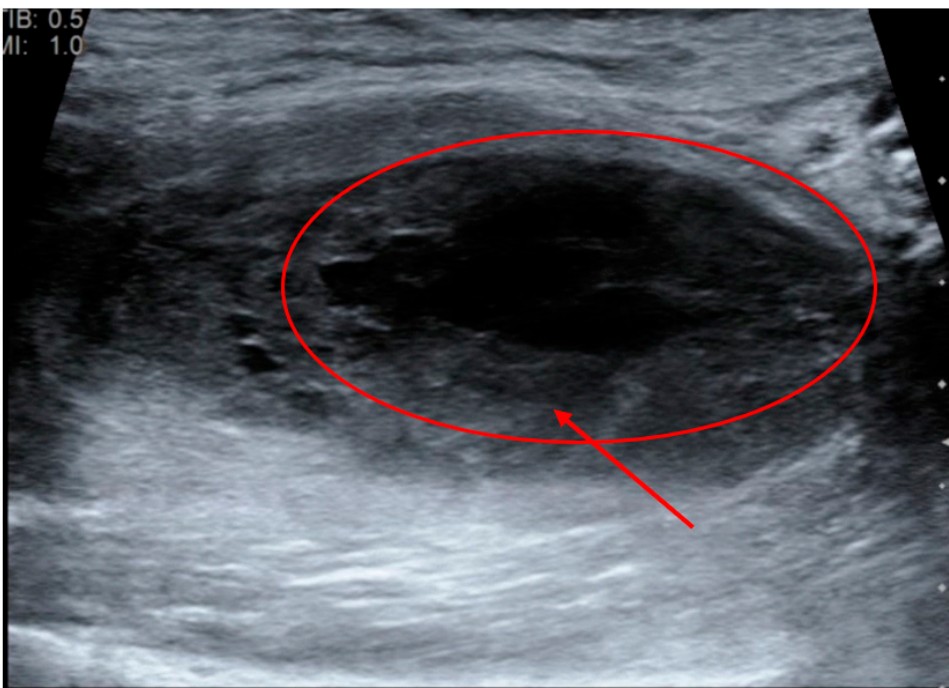

**Figure 1.** Patient presenting with multiple complaints including numbness predominantly in the median nerve distribution, inability to flex the second through fifth digits completely and inability to flex completely at the elbow. Ultrasonography identified a large hematoma in the arm after the patient had phlebotomy while on anticoagulants. The hematoma is directly adjacent to the anterior interosseous branch of the median nerve causing compression and the clinical symptoms observed. Within the figure, the circle identifies the hematoma and the arrow identifies the anterior interosseous nerve.

### 2.1.6. Diagnosis and Physical Examination

Examination of patients with AINS will demonstrate motor weakness in FPL and FDP of the index and middle fingers. The presence of a sensory deficit rules out an isolated AIN, although combination of AINS with another compressive neuropathy is still possible, particularly in cases of upper extremity trauma. Patients with AINS may exhibit a positive Tinel sign if provoked over the anterior interosseous nerve. The "Kiloh-Nevin Sign" can be seen in those with AIN, which is the inability to make the 'ok' sign with the thumb and index fingers, specifically the inability to flex the first interphalangeal joint and second

distal interphalangeal joints [35]. The pinch maneuver may aid in diagnosis of AIN [35]. This test is performed by asking the patient to pinch the fingertips of the thumb and index finger together. For those with AIN, weakness of the FPL and FDP1 will be compensated for by use of the intrinsic muscles which are innervated by the ulnar nerve [35]. EMG may demonstrate sharp waves, fibrillations, and abnormal latencies across the three affected muscles [35].

### 2.1.7. Treatment

There is no definitive treatment pathway for AINS, as debate exists in the literature [38–42]. Initial treatment of AINS is usually non-operative, as many cases are observed to spontaneously resolve over time [38,43,44]. Early administration of systemic corticosteroids and antiviral medications, such as acyclovir, have been shown to benefit patients in the acute setting but lack strong evidence to support their use [45,46]. If non-operative management fails, surgical exploration and decompression may be considered. The surgical approach for AINS involves identifying the nerve and releasing all areas of potential compression. Potential areas of entrapment include the lacertus fibrousus, tendinous edge of the humeral head of the pronator teres (most common cause), the proximal fascial margin of the FDS arch, the Gantzer muscle, or other aberrant muscle anatomy if present.

### *2.2. Pronator Syndrome*
### 2.2.1. Etiology

Pronator syndrome (PS) is characterized by volar forearm pain with paresthesia in the median nerve sensory distribution. PS is caused by compression of the median nerve between the two heads of the pronator teres muscles or at the level of the proximal arch of the FDS. It is a rare diagnosis and sufficient information does not exist to postulate its prevalence or incidence (Figure 2) [35]. Patients may present similar to CTS, with numbness or paresthesia of the radial three and half digits and pain in the volar forearm and/or wrist. PS can be differentiated from CTS by the presence of numbness or paresthesia in the distribution of the palmar cutaneous branch of the median nerve which supplies sensation over the thenar eminence. Thenar sensory symptoms are not present in CTS as the palmar cutaneous branch of the median nerve enters the hand outside of the carpal tunnel.

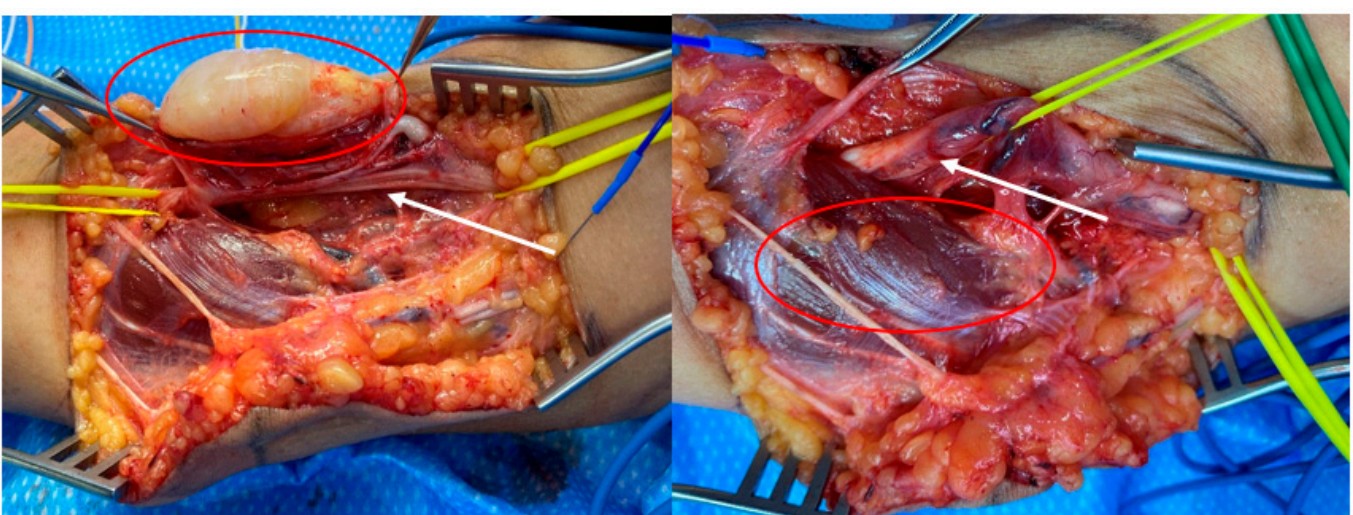

**Figure 2.** Neurofibroma causing nerve compression of the median nerve leading to pronator syndrome. Within the figure on the **left**, the circle identifies the neurofibroma and the arrow points to the median nerve. In the **right** figure, the circle identifies the pronator teres muscle and the arrow indicates the median nerve.

### 2.2.2. Diagnosis and Physical Examination

A differentiating factor between PS and CTS is the lack of nocturnal symptoms in patients suffering from PS. Additionally, Phelan's maneuver and Tinel sign should not elicit symptoms in these patients. Three maneuvers have been described to help establish a diagnosis of PS. The most common exam finding is pain and paresthesia within 30 s of compression on the proximal volar forearm on or just proximal to the pronator teres muscle belly [35]. Resisted pronation and supination may help distinguish if median nerve compression is caused by the pronator teres or lacertus fibrosus. Resisted flexion of the middle finger PIP joint can be useful as the median nerve can be compressed by the heads of FDS, however this test can also be positive in CTS. EMG studies are typically normal in PS and may have limited utility in diagnosis of this syndrome but may be helpful in ruling out other compressive neuropathies [35].

### 2.2.3. Treatment

As with other compressive neuropathies, non-surgical management is first-line treatment. Stretching the flexor/volar forearm compartment, nerve gliding exercises, and rotational immobilization of the forearm may improve the patients' symptoms. Surgical treatment is aimed at nerve decompression at the site(s) of entrapment in the proximal forearm and can be performed from multiple approaches. The benefit of surgery is not well understood, as there is limited retrospective data determining efficacy. Published studies have demonstrated only a partial relief of symptoms with surgery [35,47,48].

### *2.3. Ligament of Struthers Syndrome*
Etiology

The Ligament of Struthers is a rare anatomical accessory fibrous band of tissue that can cause compression of the median nerve and brachial artery, resulting in "Struthers Syndrome." This structure spans from the supracondylar process of the humerus to the medial humeral epicondyle [49]. John Struther first described the ligament in 1848, describing the ligament as a 'peculiar process,' and believed it to be a vestigial structure supporting his theory of evolution which was later developed and popularized by Charles Darwin in The Descent of Man. This syndrome has largely been described in case reports and usually presents with symptoms of pain, weakness, and sensory deficits [50,51]. Patient's symptoms may improve with operative release of the ligament [50–52].

### **3. Ulnar Nerve**
### *3.1. Cubital Tunnel Syndrome*
### 3.1.1. Etiology

Cubital tunnel syndrome is the second most common compressive neuropathy of the upper extremity and results to compression of the ulnar nerve at the elbow [53]. An overview of the compressive neuropathies of the ulnar nerve is provided in Table 4. Patients facing cubital tunnel syndrome are 4 times more likely to present with advanced disease than those with CTS, and if left untreated, patients can develop irreversible motor and sensory deficits, and ultimately develop significant joint contractures [53].

Table 4. Summary of ulnar nerve compressive neuropathies.

| Ulnar Nerve | |
| --- | --- |
| **Cubital Tunnel Syndrome** | |
| Compression Site | Compression within cubital tunnel, most often at level of Osborne's ligament. |
| Sensation Deficit | Dorsal ulnar hand. |
| Symptoms | Grip weakness of ring and small finger FDP. |
| Diagnosis | Positive Froment's and Wartenberg sign. EMG shows denervation of ulnar innervated muscles with prolonged latencies in cubital tunnel. |
| Complications | Injury to the medial antebrachial cutaneous nerve is common, causing remnant paresthesia and pain over olecranon. |
| **Guyon's Canal** | |
| Compression Site | Compression at Guyon's canal which lies between the volar carpal ligament and the transverse carpal ligament. |
| Sensation Deficit | Small finger and ulnar half of ring finger. |
| Symptoms | Dependent on compression zone. Zone 1: paresthesia and intrinsic muscle deficits. Zone 2: only motor deficits. Zone 3: only sensory deficits. |
| Diagnosis | First interosseous atrophy, inability to cross fingers, positive Wartenberg, Duchenne, and Jeanne sign. Thorough social and occupational history. EMG demonstrates prolonged latencies at the wrist. |

### 3.1.2. Anatomy

Comprehensive understanding of the anatomy of the cubital tunnel is critical to understanding how to address the pathology. Osborne's ligament forms the roof of the cubital tunnel and extends from the medial epicondyle and humeral head of the flexor carpi ulnaris (FCU) to the olecranon and the ulnar head of the FCU muscle. The medial/ulnar collateral ligament, medial elbow joint capsule, and olecranon make up the floor. Most commonly, the ulnar nerve is compressed at the level of Osborne's ligament. Compression can also occur proximally by the arcade of Struthers, the medial epicondyle, distally by the deep flexor pronator aponeurosis, and may be due to hypertrophic anconeus muscles (Figure 3) [53,54]. Damage to the medial antebrachial cutaneous nerve is a known complication of operative release and may lead to residual paresthesia and pain over the olecranon [53].

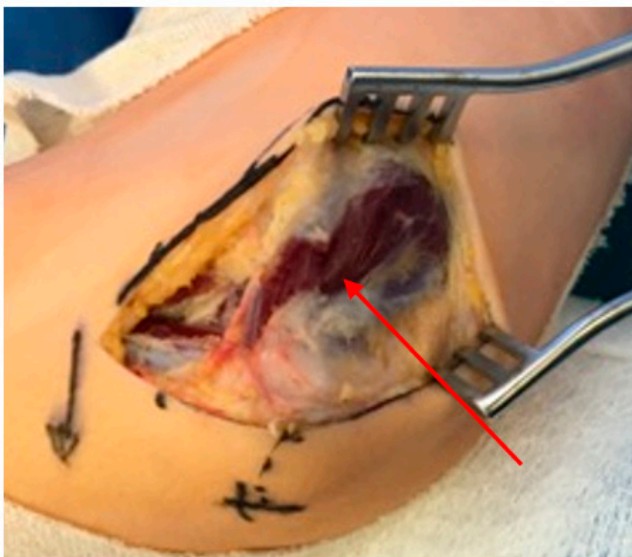 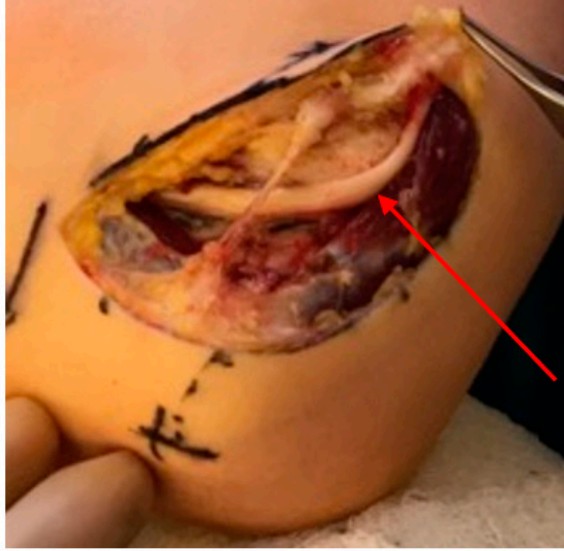

**Figure 3.** Compression of the ulnar nerve in the cubital tunnel causing cubital tunnel syndrome secondary to the anconeus muscle. Release of this muscle allows for nerve decompression and resolution of symptoms. The arrow in the image on the **left** indicates the anconeus muscle. This muscle is resected to reveal the ulnar nerve (indicated by the arrow on the **right** image).

### 3.1.3. Diagnosis

Several key physical exam findings aid in the diagnosis of cubital tunnel syndrome. Dorsal ulnar hand numbness is indicative of ulnar nerve compression proximal to Guyon's canal, as due to the dorsal cutaneous nerve branches proximal to the ulnar nerve entering the canal. Patients may experience weakness of the ring and small finger FDP and may present with grip weakness [53]. Decreased grip strength and atrophy of the intrinsic hand muscles may be present in patients with CTS. A positive Tinel sign elicited with tapping over the elbow may indicate CTS as well [55]. Patient's may have a positive Froment's sign in which they are unable to maintain a pinch grip between their thumb and index finger due to inability to adduct their thumb, and in turn flexes the thumb IP joint. Patient's may also have a positive Wartenberg sign in which the small finger is held in abduction due to extensor digiti minimi abducting in compensation for the weak third dorsal interosseous muscle [53]. Additionally, EMG studies may reveal denervation of ulnar-innervated muscles and demonstrate prolonged latencies across the cubital tunnel.

### 3.1.4. Treatment

Non-operative treatment consists of activity modification including discontinuing strenuous elbow extension movements, avoidance of pressure to the posteromedial aspect of the elbow, maintaining the elbow between 45–50 degrees of flexion, and a nighttime elbow orthosis. A 1988 review of 50 studies found that nearly 60% of patients with mild cubital tunnel syndrome found relief with nonsurgical management [56]. Surgical interventions include nerve decompression, medial epicondylectomy, and transposition of the ulnar nerve in a subcutaneous, intramuscular, or submuscular plane [53]. A 2016 Cochrane review of randomized prospective study suggested that ulnar nerve decompression may be a superior operative intervention as no differences in outcomes were observed between different procedures [57]. For those with advanced ulnar nerve motor neuropathy, an anterior interosseous nerve to ulnar motor nerve transfer can restore function of the intrinsic hand muscles [58].

### *3.2. Guyon's Canal*

### 3.2.1. Etiology

Guyon's Canal Syndrome, or ulnar tunnel syndrome, occurs with compression as the ulnar nerve passes through Guyon's canal at the wrist. Guyon's canal lies between the volar carpal ligament and the transverse carpal ligament. Sensory deficits are limited to the small finger and ulnar half of the ring finger as the dorsal and palmar cutaneous branches of the ulnar nerve branch proximally to Guyon's canal. Patients with Guyon' Canal syndrome should have no sensory deficits to the dorsal and volar hand.

Classification of symptoms can aid in diagnosis. In zone 1, patients will experience both paresthesia and intrinsic muscle deficits as the sensory and motors branches have not yet bifurcated. In zone 2, patients will exhibit only motor deficits, as only the motor fibers are compressed. Compression in zone 3 exhibits purely sensory deficits.

### 3.2.2. Diagnosis

Interossei atrophy or wasting, especially over the first dorsal interosseous muscle, inability to cross fingers, and Wartenberg sign are physical exam findings indicative of motor branch involvement. An ulnar claw deformity known as the Duchenne sign may be observed secondary to lumbrical paralysis of the little and ring fingers [59]. The Jeanne sign is representative of hyperextension of the thumb at the MCP joint secondary to paralysis of the adductor policus [59]. Evaluation of patients with suspected nerve compression at the Guyon canal should include Allen test for arterial perfusion of the hand to evaluate for potential vascular etiologies and tenderness about the carpus which may suggest fracture of the hamate or pisiform [59]. Ganglion cysts are a common cause of compression in nontraumatic patients [59]. A thorough occupational and recreational history is important in patients undergoing workup for suspected Guyon's Canal Syndrome, as pressure

from bicycle handlebar or occupational tasks requiring repetitive blunt force, such as jackhammering, are classic causes of compression at Guyon's canal. EMG demonstrating prolonged latencies at the wrist can support this diagnosis [59].

### 3.2.3. Treatment

Traditional conservative treatment may be beneficial. For those with a known etiology of compression such as a ganglion cyst, treatment of the source of compression may prove therapeutic. In severe cases or in those refractory to non-operative management, surgical decompression of Guyon's canal can be considered. In this procedure, all three zones of the canal should be addressed, including the principal compressive structures of the antebrachial fascia, volar carpal ligament, and hypothenar fibrous arch. Surgeons must note the proximity of Guyon's canal and the carpal tunnel, as the Guyon canal may be inadvertently released during carpal tunnel release in an ulnar deviated incision. Patients experiencing both ulnar and median symptoms at the wrist, may benefit from simultaneous release of the two canals through a single incision.

## 4. Radial Nerve

### *4.1. Radial Tunnel Syndrome*

### 4.1.1. Etiology and Anatomy

Radial tunnel syndrome is thought to be caused by compression of the posterior interosseous nerve in the proximal forearm, however the existence of this syndrome is questioned in the literature [60,61]. The radial tunnel is a potential space approximately 5 cm from the level of the radiocapitellar joint which extends past the proximal edge of the supinator. It is laterally bound by the brachioradialis, extensor carpi radialis longus, and extensor carpi radialis brevis muscles. The medial bounds are the biceps tendon and brachialis, and the deep surface is the radiocapitellar joint. The radial nerve splits into the radial sensory nerve and the posterior interosseous nerve proximal to the supinator at the elbow joint. An overview of radial tunnel syndrome and Wartenberg syndrome are provided in Table 5.

The posterior interosseous nerve is the terminal motor branch of the radial nerve and can be compressed at several sites. Potential sites of compression include the aponeurotic edge of the supinator (the arcade of Frohse), the medial edge of the extensor carpi radialis brevis, the recurrent branches of the radial artery (Leash of Henry), and the inferior margin of the superficial layer of the supinator muscle. Most often, compression occurs at the Arcade of Frohse [60].

### 4.1.2. Diagnosis

Presenting symptoms include pain along the dorsoradial aspect of the proximal forearm with focal pain 3–5 cm distal to the lateral epicondyle. Pain may radiate proximally and distally and may be exacerbated by pronation and supination of the forearm. Patients may have motor weakness, and no sensory deficits are typically observed. Occupations with recurrent supination and pronation of the forearm and resisted extension of the elbow may face increased rates of radial tunnel syndrome [62,63]. Clinicians should include lateral epicondylitis on their differential due to its increased prevalence and location. Differentiation of the two syndromes can be made by performing wrist extension against resistance. Patients with lateral epicondylitis will have increased pain with this testing maneuver while patients with radial tunnel syndrome usually do not. Radiographic evaluation is typically nondiagnostic for Radial Tunnel Syndrome, however studies have suggested that MRI can reveal soft tissue changes in these patients [64]. EMG studies are typically not useful [64].

### 4.1.3. Treatment

Non-operative management is first line and includes avoidance of maneuvers involving prolonged elbow extension and forearm pronation and wrist extension. Steroid

injection has been found to benefit in these patients as 72% of patients undergoing injection reported symptom relief at 6 weeks and 62% had relief at 2 years [65]. Surgical intervention may be attempted if conservative treatment fails. Several surgical techniques have been described without clear support for a superior method, and all techniques appear to achieve satisfactory results between 70–90% of cases [66,67]. Identification of the posterior interosseous nerve with proximal and distal exploration are crucial in all surgical approaches (Figure 4).

**Table 5.** Summary of radial nerve compressive neuropathies and physical examination techniques across all nerves.

| Radial Nerve | |
|---|---|
| **Radial Tunnel Syndrome** | |
| **Compression Site** | Proposed compression of posterior interosseous nerve in proximal forearm, with multiple potential sites for compression. |
| **Sensation Deficit** | No sensory deficits. |
| **Symptoms** | Pain along dorsoradial proximal forearm, may worsen with pronation and supination. |
| **Diagnosis** | Thorough social and occupational history. MRI may demonstrate soft tissue changes. |
| **Wartenberg Syndrome** | |
| Compression of radial sensory nerve in forearm between fascial layers of the brachioradialis and extensor carpi radialis longus. May cause pain in dorsum of thumb, index finger, and radial half of middle finger. | |
| **Physcial Examination Techniques** | |
| **Tinel Test** | Elicted through tapping over the flexor retinaculum which creates paresthesias. Indicative of carpal tunnel syndrome (sensitivity 62%, specificity 93%) [68]. Typically negative in pronator syndrome. |
| **Phalen Test** | Produced through 90 degree wrist flexion with resultant paresthesias. Indicative of carpal tunnel syndrome (sensitivity 85%, specificity 90%) [68]. Typically negative in pronator syndrome. |
| **Kiloh-Nevin Test** | Evident through inability to make 'ok' sign with thumb and index finger. Indicative of anterior interosseous syndrome. |
| **Froment's Sign** | Inability to maintain pinch grip, indicative of cubital tunnel syndrome. |

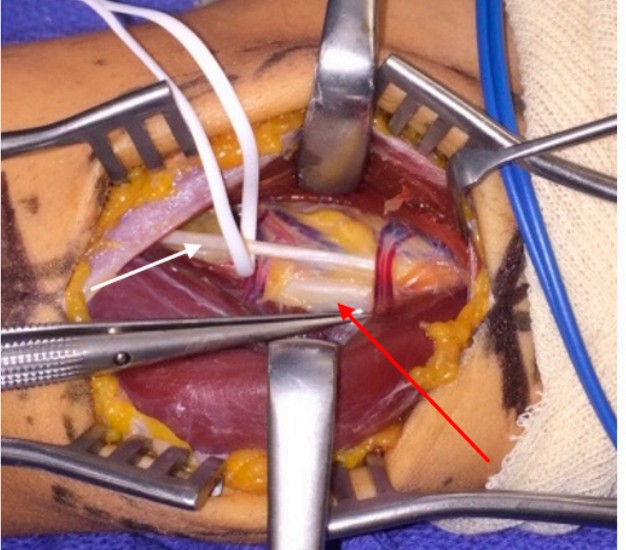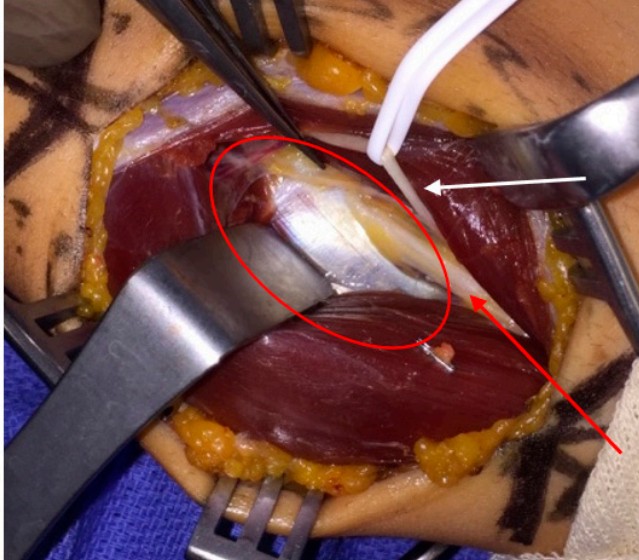

**Figure 4.** Identification of the posterior interosseous nerve (red arrows) deep to the extensor carpi radialis brevis during release of the posterior interosseous nerve for radial tunnel syndrome. The sensory branch of the radial nerve is observed branching proximal to the radial tunnel (white arrows). The circle in the right image encompasses the supinator fascia which is released to decompress the radial tunnel.

### 4.1.4. Wartenberg Syndrome

Compression of the radial sensory nerve in the forearm was first described by Wartenberg as "cherialgia paresthetica" [69]. The radial sensory nerve provides sensation to the dorsum of the thumb, index, and radial half of the long finger and compression causes pain in this distribution. It is superficially located and is susceptible to external compression during pronation from the fascial layer between the brachioradialis and extensor carpi radialis longus. Wartenberg syndrome has been described to have been caused by a variety of etiologies including a fascial band and lipoma compression [69,70]. Treatment is non-operative at first with progression to surgery as needed. Surgical decompression in Wartenberg syndrome has less favorable results when compared surgical outcomes for other upper extremity compressive neuropathies [71].

## 5. Limitations

This article has limitations and was a literature review. Strict inclusion and exclusion criteria were not utilized when identifying appropriate articles. Additionally, quality of the literature search was not assessed.

## 6. Conclusions

A comprehensive patient history and physical examination can help identify potential sites of nerve compression. Each syndrome's treatment pathway begins with non-operative measures, with escalation to surgery if needed. Carpal tunnel syndrome is the most common upper extremity nerve compression syndromes and surgical release is therapeutic, with similar outcomes between open and endoscopic approaches. Other compression syndromes are less common but warrant appropriate investigation if suspected.

**Author Contributions:** Review of literature and composition of ideas, C.J.B. and S.S. Writing of the manuscript, C.J.B., N.P.S. and J.X.R. Review and approval of the final version for submission, C.J.B., N.P.S., J.X.R. and S.S. All authors have read and agreed to the published version of the manuscript.

**Funding:** This research received no external funding.

**Institutional Review Board Statement:** Approval was not required for this review.

**Informed Consent Statement:** Informed consent was not obtained for this study.

**Data Availability Statement:** Not applicable.

**Conflicts of Interest:** The authors declare no conflict of interest.

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
