# Peer review of "Compression Neuropathies of the Upper Extremity: A Review"

_2673-4095, doi:10.3390/surgeries2030032_

Round 1
Reviewer 1 Report
The manuscript entitled “Compression Neuropathies of the Upper Extremity: A Review” is a narrative review study describing the most common compression neuropathies of the forearm and their management.
The manuscript provides an overview of the most common neuropathies encountered in clinical practice. In my opinion, the topic is interesting but it needs a more comprehensive analysis due to the wide topic chosen by authors (clinical features, diagnosis and treatment).
The present form lacks the aim to enlarge the current knowledge on the topic, and it traces information well-known. In my opinion, it is necessary to better focus the research question on a specific aspect of the compression neuropathies (surgery?).
Following, it is reported a point-by-point revision by section.
TITLE
The main idea is clearly stated. As introduced above, the topic is too wide for the current form of the paper. It lacks the research question.
ABSTRACT
The contents of the abstract mirror the content of the full paper however the abstract’s layout (introduction, methods, results, conclusions) does not match the one of the text where only “introduction” and “conclusions” are clearly stated.
INTRODUCTION
The text is correctly written but the purpose of the study is not clearly stated. In the present form, the paper is too scholastic and it is not clearly analyzed in terms of a scientific research question.
METHODS
The process of selection or exclusion of the articles is not stated.
The paper does not analyze the ultrasound diagnostic assessment.
RESULTS
A true “results” section is missing.
Tables 1,2 and 4 include the main information. Table 3 only includes generic statements; I suggest adding some percentages or values to represent the costs, if available.
Page 7, lines 7-8, it seems that something is missing here. The sentence should probably be rephrased as the following: “Stretching the flexor/volar forearm compartment, nerve gliding exercises, and rotational immobilization of the forearm may improve the patients’ symptoms”.
Regarding the classification of symptoms in Guyon’s canal syndrome (page 9, line 12-15), I suggest adding a figure to facilitate the reader in understanding the anatomy of Guyon’s canal.
DISCUSSION
The discussion section is missing.
Limitations are not noted.
CONCLUSIONS
The conclusions capture the essence of the analyses provided however do not add any new relevant insight on the workup and management of compression neuropathies of the forearm.
FORM, STYLE, AND SUBSTANCE
The manuscript is clear and well written however, the style of the tables is a bit basic.
Figure 1 is out of proportion and reading the description while looking at the picture is difficult. In figure 3, I suggest adding something to highlight the subject of the picture.
REFERENCES
A reasonable selection of references was used but they are not fairly up-to-date.
Reviewer 2 Report
In this article, the authors describe the clinical characteristics of the principal compressive neuropathies. The structure of the article is nice, the pathologies are appropriately described, and in general the manuscript is well written. The figures are high-quality, but the description of the anatomical structures can be improved (see comments below).
I believe this article represents a good overview of the compressive neuropathies, which can be useful for clinicians and physiotherapists. I hope my comments below could help improve the quality of this manuscript.
- “Compressive neuropathies of the forearm are commonly encountered by cli- nicians and within we provide an overview of commonly encountered neuropathies.” I believe this sentence in the abstract has some missing words.
- In table 1, the authors mention “Optimization of comorbid conditions”. It would be great if the authors could clarify whether there are some specific comorbid conditions whose optimization would be particularly beneficial to the patients.
- In table 2, the authors report that Carpal Tunnel Syndrome may lead to thenar atrophy. They define this as a symptom, but it should be noted that this is not a symptom, but a finding of the physical examination.
- “Electrophysiological assessment through EMG are highly sensitive in detecting CTS” It would be of interest to the reader to report the EMG findings which help in the diagnosis of CTS.
- “Schrier et al. found of patients undergoing release surgery for CTS found that patients reporting more satisfactory experiences with their healthcare provider and health system also reported better outcomes following surgery.” I believe this sentence has some errors: “found” is repeated twice. I suggest to rephrase.
- In figure 1, it would be great to clarify with arrows and/or circles where the hematoma and the anterior interosseous branch of the median nerve are.
- Figure 1 is not contextualized in the manuscript.
- “lacterus fibrosus” should be “lacertus fibrosus”.
- “PIP join” should be “PIP joint”.
- In figure 2, it would be useful to indicate the anatomical structures relevant to the purpose of the figure.
- “and rotational immobilization of the forearm may patients' symptoms” I suppose there is a missing verb in this sentence.
- In figure 3, it would be useful to indicate the anconeus muscle and the ulnar nerve if visible.
- In figure 4, it would be useful to indicate the anatomical structures relevant to the purpose of the figure.
- In table 5, there’s a typo: “extensor carpi radilais longus” should be “extensor carpi radialis longus”
Reviewer 3 Report
In this article, a narrative review on compression neuropathies of the upper limb is reported. This article offers an interesting and complete overview on the topic. Below my comments.
Although this article is a narrative review, the authors should provide information on how they did the literature search, which articles they decided to use for their narrative review.
More details should be provided on the tests (e.g., Tinnel test) for each compression neuropathies that can be used in the diagnosis; the reader will surely appreciate detailed information on what the procedures of these tests are, how to interpret them and information on their reliability and validity. Reporting this information in a table would probably improve the scientific quality of the manuscript
More information on conservative treatments (e.g., physiotherapy) may be useful to the reader
Epidemiological data (e.g. prevalence, incidence) for each compression neuropathies are not reported.
The authors should make homogeneous the sub-paragraph subdivision for each compression neuropathies: I suggest organizing the sections according to this order: Introduction (with epidemiological data), etiology, assessment and diagnosis, treatment
A section of the study limitation is missing. In this section the authors should emphasize that this study design is a narrative review with all the limitations that goes with it.
Round 2
Reviewer 1 Report
The design of the study is still flawed and the obtained results are questionable. This could be due to the used authors' biased selection of Boolean key terms. The corrections brought to the initial paper are not sufficient in my opinion to justify a possible publication, at the light of the reason adopted by the authors in their response to the reviewer. The manuscript remains too scholastic not adding anything to the present research on the topic.
Reviewer 3 Report
The authors have fully replied to all my comments. In my opinion the manuscript is now ready for its publication
